# A Medical Image Visualization Technique Assisted with AI-Based Haptic Feedback for Robotic Surgery and Healthcare

**Georgios M. Minopoulos** [ID]**, Vasileios A. Memos** [ID]**, Konstantinos D. Stergiou** [ID]**, Christos L. Stergiou** *[ID] **and Konstantinos E. Psannis** *[ID]

Department of Applied Informatics, University of Macedonia, 54636 Thessaloniki, Greece
* Correspondence: c.stergiou@uom.edu.gr (C.L.S.); kpsannis@uom.edu.gr (K.E.P.); Tel.: +30-2310-891-737 (C.L.S.)

**Abstract:** A lesson learned during the pandemic is that social distancing saves lives. As it was shown recently, the healthcare industry is structured in a way that cannot protect medical staff from possible infectious diseases, such as COVID-19. Today's healthcare services seem anachronistic and not convenient for both doctors and patients. Although there have been several advances in recent years, especially in developed countries, the need for a holistic change is imperative. Evidently, future technologies should be introduced in the health sector, where Virtual Reality, Augmented Reality, Artificial Intelligence, and Tactile Internet can have vast applications. Thus, the healthcare industry could take advantage of the great evolution of pervasive computing. In this paper, we point out the challenges from the current visualization techniques and present a novel visualization technique assisted with haptics which is enhanced with artificial intelligent algorithms in order to offer remote patient examination and treatment through robotics. Such an approach provides a more detailed method of medical image data visualization and eliminates the possibility of diseases spreading, while reducing the workload of the medical staff.

**Keywords:** 5G; artificial intelligence; augmented reality; haptics; tactile internet; virtual reality

## 1. Introduction

Nowadays, the healthcare industry seems to need the assistance of technology in order to become more efficient now more than ever, as the entrepreneur and healthcare investor of the Silicon Valley, Vinod Khosla claimed in late 2016 "In the next ten years, data sciences and software will do more for medicine than all of the biological sciences together" [1]. Telemedicine, widely known as e-Health, will boost the evolution of the healthcare industry to include almost the whole population, particularly in low-income and remote areas, where patients will have the opportunity to monitor their personal vitals with considerable convenience by using low-cost devices and to utilize technologies such as remote diagnosis and treatment. Innovative solutions could be also deployed in medical imaging data. The equipment used for medical imaging are able to produce both 2D images, such as radiographs, and 3D volumetric image datasets, such as Magnetic Resonance Imaging (MRI), Computed Tomography (CT), and Positron Emission Tomography (PET). Existing methods of medical data visualization comprise the typical depicting of volumetric data, which is a part-by-part viewing technique for coronal, sagittal, and axial imaging planes, or corresponding oblique reformats. These limitations can be minimized using depth 3D imaging that Virtual Reality (VR) and Augmented Reality (AR) are able to provide. VR and AR technologies signify a momentous progress for medical staff, by abolishing their imagination capabilities as an imperative prerequisite for the interpretation of medical images.

Thanks to another emerging technology that provides remote touch, known as haptics, it is possible for doctors, surgeons, and medical staff to access, control, and depict the 3D

imaging data of patients from a distance in a similar or better way than the regular distance. Thus, they are able to have a better clinical picture, detecting potential complications occurred by a disorder or infectious disease, such as COVID-19. In addition, including Artificial Intelligence (AI) to the system, it is able to "learn" the controller's actions during the operation. Thus, it will be able to perform operating procedures via minor or entirely no human monitoring or manipulation. The latency between telemedicine applications and master control devices is one of the most significant technical issues that 5G mobile networks are expected to answer. The main aim of 5G systems is to keep the latency of wireless communications below acceptable thresholds, as the use of robotic surgical machines can be manipulated distantly by proficient surgeons, who may be located many kilometers far away from the patient. 5G networks are necessary to deliver essential communication stability and to guarantee minimum latency for the successful transmission of haptic data. Moreover, better-quality visualization and increased dexterity will be provided through 5G networks' improved data rates, resulting in a higher-precision surgical process.

The combination of emerging technologies is very meaningful as it can boost the medical sector. Well trained medical staff that are capable of using such innovative solutions benefit both them and the patients. An improved comprehension of the patient's medical status could provide better treatment, which may be vital for their life. Moreover, by using haptic devices, the medical staff is not obligated to come in contact with the patients, thus limiting disease spread in the case of patients who suffer from an infectious one.

As a result, the presentation of the basic theoretical information of the research field of this work provides the following main contributions:

- The role of combining emerging technologies to assist the healthcare industry.
- A novel system that introduces medical image visualization and potentially autonomous robotic surgery.

The rest of the paper is organized as follows: Section 2 presents the current work based on haptics and visualization techniques. Section 3 describes the current methods of imaging medical data and their challenges. Section 4 explains the involved technologies. Section 5 analyzes the proposed technique and provides its benefits for future exploitation. Finally, Section 6 concludes the paper and gives some potential future directions.

## 2. Related Work

An assessment of the utility of Mixed Reality (MR) in medical education is presented in [2]. This study examines the pedagogical value of virtual reality methods in medical education. The authors surveyed 258 people, who were staff of a medical college, regarding potential applications of MR in the medical curriculum. They asked them to complete a questionnaire containing eight questions. The results showed that most of the volunteers think that MR-supplemented education is advantageous over classical instruction. The 3D visualization capabilities of MR, especially in anatomy classes, were most highly valued. A valuable conclusion extracted from this study is that MR expands the capabilities and effectiveness of remote learning, which was normalized during the COVID-19 pandemic, to ensure effective student and patient education. MR-based lessons, or even select modules, provide a unique opportunity to exchange experiences inside and outside the medical community.

In [3], a technique for an orthopedic surgical process is proposed, with the establishment of the Virtual Surgical Environment (VSE) for educational purposes of the medical staff. The proposed method is used to tackle fractures of the femur and is called Less Invasive Stabilization System (LISS) surgery. The entire approach involved first getting the appropriate knowledge of the LISS plating procedure via interactions with proficient orthopedic surgeons and utilizing information-centric models. These models present a well-defined base to design and develop the emulator. A haptic interface is incorporated to support training activities in order to allow users to feel, touch, and interact with the surgical tools through these trainings.

The study in [4] demonstrates the research progress and developments in Virtual Reality (VR) and Augmented Reality (AR) technologies in the manufacturing industry. During the fourth revolution, the penetration of new immersive technologies is of main importance and focuses especially on human machine interaction. However, critical challenges in VR/AR technology implementation at a hardware and software level still exist.

## 3. Limitations of Conventional Medical Methods

It is a fact that a thorough pre-operative plan can be more effective in terms of reduced surgery operation time and fewer possibilities of morbidness. Nevertheless, current pre-operative plans are not so deep, since they are mainly based on complex 2D graphical interfaces created by the surgeon, instead of a surface visualization. The traditional view of volumetric health data affects the overload due to the great volume of the generated datasets. Although small lesions can be detected by taking a slice-by-slice analysis, this method is time-consuming. Besides, an intellectual construction of a 3D image, reviewing slices is regarded as a big challenge [5]. Well-known medical applications such as CT and MRI make use of data volume visualization. The sequential generated images are translucid and blurred, making the data volume realization difficult [6]. Moreover, surgeons use their sense of touch during an operation very much. Conversely, surgical plan systems do not use the sense of touch to supplement the visual interface. As a result, in some cases, a general surgery plan may be overloaded and take a lot of time to find the best surgical strategy [2].

Common practices of surgical training incorporate exercise using cadavers, animals, and synthetic mockups. The drawbacks of these old-fashioned approaches are already known. Animal rights' activists disapprove of the experimentation of surgical pieces of training on animals. Furthermore, the use of cadavers includes the likelihood of risk of infections, while synthetic mockups are costly and are not patient specific. Other methods entangle amateur medical staff observing the surgery executed by a professional surgeon and then gradually succeed in aiding in surgeries. The introduction of emerging technologies in the healthcare industry can eliminate obsolete methods and boost its development [2].

## 4. Proposed Approach

There are many challenges of applying innovative technologies in medicine, especially for tackling epidemic diseases. Thus, an effective scheme that involves cutting edge technologies is imperative to achieve this goal. Our proposed model is the integration of the following emerging technologies, which can constitute a robust system that will show improved detection of disorders and surgical operating capabilities.

The usage of VR and AR technologies and with the progress of human–computer interaction approaches are able to retrieve volumetric data in a 3D environment. Three-dimensional images give a more detailed anatomical description of the human body than "old-fashioned" 2D images. By wearing VR/AR glasses, doctors can obtain specific information about a patient's condition. Then, due to haptic technology, a surgeon could operate a surgery via a haptic device without being physically in the same room with the patient. Through 5G networks, the surgeon could manipulate a robotic device wherever it is located, which interacts simultaneously and precisely with the surgeon's movement. By embedding AI between the haptic device and the robotic machine, the system could be able to potentially operate independently by learning from surgeon's actions.

### 4.1. Virtual Reality (VR)

The usage of Virtual Reality (VR) in the area of healthcare has developed rapidly in recent years, enabling 3D visualization from many angles, without covering details, as is the case with 2D models. In our proposed scheme, VR can help rehabilitate patients with mobility problems, help reduce pain, treat phobias and post-traumatic stress, study various diseases in the study of viruses, and also assist in vaccine research. VR can also be used in

medical science, in the training of doctors and other specialties related to health care, as well as in the execution of medical procedures.

In addition, our proposed scheme can be integrated into other state-of-the-art systems used in robotic surgery and other necessary medical activities. For example, the use of VR and robotic surgery with the Da Vinci system can provide the possibility of a complete and integrated image of the patient before the operation and during it, diminishing the possibility of medical errors. In addition, virtual reality methods such as the "Head Mounted Display" device, CAREN (Computer Assisted Rehabilitation Environment), and Kinect programs, the Shark Punch game, various types of exoskeletons, and other media can also be used to rehabilitate patients with mobility, neurological, and mental problems.

As it is known from previous literature, VR can be full or semi-immersive and has been used to create an immersive and interactive environment [7]. Full-immersive VR displays a virtual image while the real environment is excluded from the projection. Semi-immersive VR displays the virtual image while the real world is partially excluded from the projection [8]. Thus, based on these two types of VR, our proposed model can allow doctors to effectively connect with patients and increase the frequency of medical contact. Additionally, VR technology harnesses the power of user interaction and immersion. This makes our system easier and more intuitive to operate. This can be very important in facilitating the diagnosis, as the doctor can move around the system and visualize it in more detail. The application of VR technologies into our system contributes to better image quality and interactions, facilitating a correct and more accurate diagnosis. It applies to careful study of areas that are considered to be of greater interest for diagnosis [9].

Moreover, VR aims to train in intensive care during the first critical hour, where the patient is delivered into the hands of nurses, which combines features developed in a real educational process with the possibility of using medical simulations. The virtual world is judged as a training tool for students and causes them to try their skills in various simulation permutations. Other relative benefits are considered to be the facilitation of distance learning and the tremendous flexibility in study hours. Since VR allows society to improve and facilitate their daily lives, virtual worlds are now widely used in the domain of healthcare and, in particular, in the training of doctors, aiming at the best use of resources and different learning options in a safe environment.

### 4.2. Augmented Reality (AR)

The limitation of VR systems in medicine stems mainly from the user's concern for the surrounding physical world as opposed to the virtual world. Augmented Reality (AR) surpasses them by providing a simple and straightforward User Interface (UI) over the electronically enhanced physical world.

Augmented capabilities can be integrated into Augmented Reality (AR) systems or Mixed Reality (MR) systems. Generally, AR refers to a virtual world in the real world and is a robust visualization tool. AR offers visualization, annotation, and storytelling, while MR merges the virtual world with the real one and is interactive, in real-time, and in 3D. In other words, MR makes use of AR and Augmented Virtuality (AV), as shown in Figure 1, and is regarded to be the most sophisticated technology.

Both AR and MR systems offer an immersive display of a virtual image and a real image simultaneously in real-time. Thus, the users have the opportunity to interact with the virtual image and the real world by using a Head Mounted Display (HMD), as to display the virtual image and the real-world image. The difference is that the virtual image is displayed as transparent similar to a hologram in AR, such as in the Meta-2 Development Kit and the former DAQRI systems, while the virtual representation is depicted as solid in MR, such as in the Microsoft Hololens headset [10].

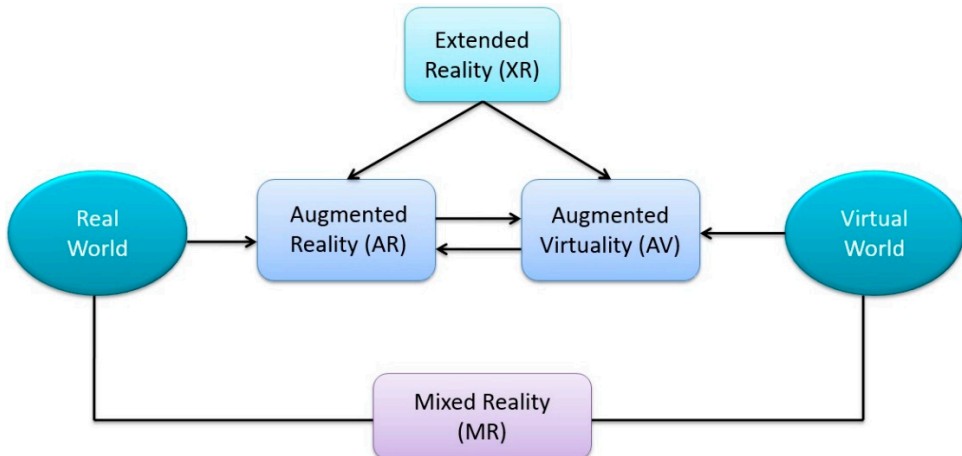

**Figure 1.** Mixed Reality continuum.

In the case that pre-operative, intraoperative, or physical examination need medical imaging, medical applications can be used for real image exportation, which corresponds to the actual body of a patient. AR is extensively used in both medical imaging and computer surgery communities. Thus, AR can be applied in different methods of body visualization such as X-ray, ultrasound, MRI, CT, and others, to offer a better quality of 3D virtual data, satisfying requirements in terms of service quality, reliability, and accuracy [11]. Based on these methods, many dangerous distortions of vital organs (i.e., spleen, liver, and lungs) caused by epidemic viruses, aneurisms, and fractured bones can also be detected by our system with high accuracy and effectiveness thanks to AR.

Generally, for any system depending on the visualization of AR, three basic requirements are essential: reliability, usability, and interoperability. Because the persuasive power of this type of visualization is very high, we can use such visualization software that can guarantee certain accuracy limitations by following it during the execution time.

The aim of visualization procedure in our proposed approach during medical procedures, such as minimally invasive surgery, is to make the medical staff feel as if they are actually in an operation room. In this way, surgeons can exploit these various kinds of wearable AR glasses to visualize vital health data from scans overspread with the patient's body while they operate. A typical pair of AR glasses used in our system has common and infrared depth-sensing cameras [12,13], microsensors such as a gyroscope and an accelerometer, and transparent lenses, which provide surgeons a greater dexterity and control, while improving their surgical accuracy.

Besides, AR enables an easy-to-use environment where the consultation and diagnosis of a 3D model produced by radiological results can be easily performed. By using this technology, our virtual system components can be placed in the real world and achieve a lighter interaction using a tactile device. This means that the surgeon can work in the operating room and interact with the patient making plain movements. The ability of the surgeon to interact with the patient in such a way is very important to avoid any kind of infection, especially in the case of completely virtual surgery [9]. Finally, AR can create new job opportunities for experienced doctors with AR knowledge and skills.

### 4.3. Computer Vision

Computer vision and VR can work together to make products more sophisticated and user responsive. Computer vision aids VR with robust vision capabilities such as simultaneous localization and mapping, structure from motion, and user body tracking and gaze tracking. Computer vision for AR enables computers to obtain, process, analyze, and understand digital videos and images. By looking at an object and its appearance, location, and settings, it identifies what the object is. In fact, computer vision goes above and beyond working closely with VR and AR to provide the users with sophisticated interactive content.

Moreover, with the use of computer vision with AR, surgeons can place surgical incisions more precisely and prevent tissue damage [14].

### 4.4. Haptics/Tactile Internet

Haptic data transmission demands an ultra-reliable and ultra-responsive connectivity of networks that is mandatory for the successful real-time interaction of actuation and touch. This type of communication is the purpose of Tactile Internet. It will transform many parts of society with exceptional applications drastically and change today's concepts of the Internet from content oriented to a delivery of haptics. Haptic communications will unveil a completely novel Internet through machine-to-machine and human-to-machine interaction, by offering high reliability, low latency, improved security, and expanded network coverage, which comprise the overwhelming conditions for real-time interactive systems. Haptic communication is a rapidly growing technology with applications in several domains ranging from education and entertainment to telerobotics and realistic simulators for planning and training of complex surgical procedures.

Haptics promises to improve surgical planning by giving doctors access to all the virtual instruments they will need for performing operations. A surgeon may, therefore, feel and touch the patient's bones, muscles, and essential organs. Additionally, contact forces help the surgeon avoid interpenetration of body areas that could be difficult to see. By using haptic feedback to simulate certain surgical operations for teaching purposes, realism is heightened [2]. For educational purposes, simulating specific surgical procedures by implementing haptic feedback will lead to increased realism. A haptic interface provides improved precision and quicker accomplishment of tasks with rarer technical errors, while a visual–haptic interface provides an advanced way to execute force sensitive tasks [15]. Thus, haptic data reinforced by low latency networks will make VR-based headsets that can assist surgeons and doctors in executing partially operations via telepresence available, providing many benefits.

The transmission of touch and grasp skills will be improved via the Tactile Internet, as illustrated in Figure 2. The utilization of such a technology will empower the physical capabilities of medical professionals to offer a precise diagnosis and treatment remotely. It is anticipated that real-time remote interaction and consultation will be enhanced by the transmission of haptic data that incorporates the sense of touch. Moreover, VR and AR can be used for virtual therapy by delivering an immersive user experience. The aforementioned unique characteristics of e-health are the main incentives that will transform the medical industry in the near future, offering many benefits and capabilities.

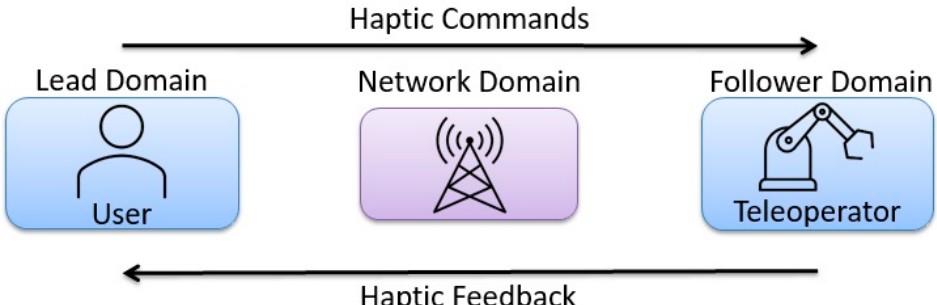

**Figure 2.** Tactile Internet communication domains.

### 4.5. Artificial Intelligence (AI)

Artificial Intelligence (AI) has revolutionized medical technology by offering possibilities with beneficial results for humans. It is an important part of computer science and is able to cope with various medical problems, finding applications in many areas that usually include large amounts of data [16]. AI models are developing significantly and help the evolution of medicine in such a way as to allow more easily the prognosis, prevention, and personalized treatment of patients in a shorter period of time. A typical example is

the use of an electronic personal health file as a measure of monitoring vital functions with biosensors, as well as a useful means to achieve optimal therapeutic compliance of patients. The key objectives are to thoroughly analyze recent developments of AI in medicine, all of the cases that can already be used in clinical practice, and the challenges and risks faced by both healthcare professionals and their respective institutions in implementing AI in the daily practice and training of future doctors.

Other applications of AI in the field of medicine are the creation of robotic systems. Initially, the development of a robotic platform appears to provide the ability through the use of ultraviolet radiation to kill viruses, bacteria, and other types of harmful microorganisms by destroying their genetic structure material [17]. This technique can contribute to limiting the spread of infectious diseases by ensuring their protection from the harmful action of microorganisms. Robotic systems are also being used even in simple daily procedures, such as taking blood from people who are patients undergoing chemotherapy or with a heavy body type [18].

Some advantages of the use of Artificial Intelligence and robotic surgery are the reduction in complications in surgical operations, the time of surgery, the patient's stay in the hospital, the time of recovery and reintegration into social activities, and the serious financial cost of insurance funds. A variety of data can be collected and used to truly empower the patients' time, facilitating improved health and wellness.

*4.6. 5G Networks*

The exceptional revolution in wireless technology aims to connect societies and professionals with excessive responsiveness. Fifth generation (5G) networks are expected to achieve a 1000 times upgrade in data rate with almost zero latency as compared to the present generation. The latency of 5G is anticipated to be up to 1 ms, i.e., 20 times improved from 4G [19]. The upcoming standard of 5G will also generate numerous pioneering health services. Consequently, the care of patients will increase in quality, while the medical staff workloads and costs will be reduced.

5G will boost the prospects for the remote educational courses of doctors and surgeons by implementing visual and haptic communication. This will create an Internet of medical skills, where medical staff will be able to transfer their expertise and share their experience overcoming the obstacle of distance by using robotic and haptic devices. Thus, a main advantage is that amateur surgeons will have the opportunity to attend and experience surgical procedures by experienced doctors, even if they are located in different places. Furthermore, robots will be manipulated by surgeons with the utilization of haptic gloves or devices, which can transfer haptic feedback to the operating doctor despite the distance. This surgical operation of the future will become feasible due to the 1 ms latency that 5G promises to offer.

An additional innovative application that 5G networks are anticipated to bring to light is bio-connectivity. This technology refers to the on-the-move delivery of medical care, which will offer medical services to patients without being transferred to hospitals. Thus, it will help to the decongestion of the hospitals. Evidently, the benefits of implementing such a technology start from improving performance in hospitals to novel methods of monitoring the patients' health and the offer of individualized medication.

The development of 5G will engage several technologies, of which some of them are already being used by present generations. In addition, it will take advantage of extremely high-frequency bands in order to deliver faster mobile broadband speed. A crucial issue of 5G technology is to become widely accepted by humanity. The smooth operation of healthcare applications demands a broadly reliable coverage of 5G networks, which is feasible with the placement of numerous antennas in specific spots. Though, some groups of skeptics believe the opinion that electromagnetic waves straightforwardly affect human lives, negatively influencing some parts of the population. Thus, research on this topic that prove the safety of 5G technology is fundamental for facilitating a variety of e-health applications [20].

## 5. Model Architecture

Medical professionals will have very powerful physical tools to visualize and work in a 3D environment by visualizing medical image data as 3D models. This technology does not only sound attractive, but it will become a necessary tool in their hands. Applying 3D images will considerably eliminate the present demand to have an imaginative capability to mentally form the anatomical structures of a 3D patient from the visualization of 2D medical images [9].

The cumulative usage of VR and AR technologies with the progress of human–computer interaction approaches contribute to retrieving volumetric data in a 3D environment. The manipulation and potential diagnosis from the 3D pictures are made easier by more advanced visualization tools and interaction approaches that VR, AR, AI, and haptics promise to give in healthcare. Numerous systems have been experimented on by using haptic feedback devices to interact with volumetric data in VR and AR settings. In spite of having the competence to interact with volumetric data in 3D conditions, these techniques utilize one-point haptic interfaces [6].

The combination of VR, AR, AI, and haptics can offer an interactive solution for surgeons to enhance their profession. Since there is limited risk of disease transmission between surgeons, they may check patients without having them physically present in the hospital or operating room. This method may be particularly helpful in circumstances when patients suffer from infectious disorders.

As depicted in Figure 3, a doctor or surgeon wears VR/AR glasses that give them the capability to examine the human body of the patient in a 3D environment and obtain a detailed description of their condition. Moreover, due to haptics, the surgeon can operate a surgical procedure by controlling a haptic device that manipulates a surgical robot or slave device on the other edge. This means that the surgeon and the patient could be in different places physically, instead of in the same operating room. Thus, potentially every surgeon can participate in surgical operations all over the world without moving from their permanent residence. Using AI between the haptic device and the slave device, the system will be able to self-learn by "observing" the surgeon's activity, giving the opportunity to become fully independent from human interaction.

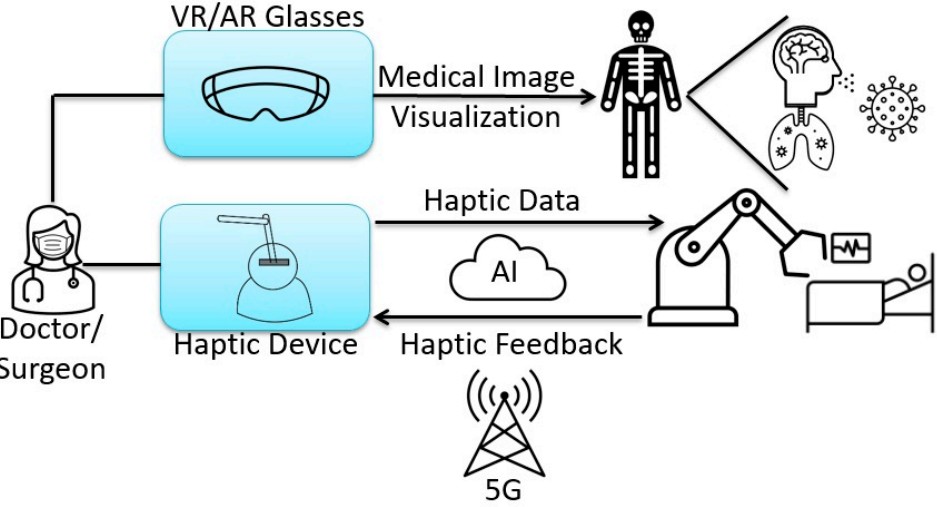

**Figure 3.** Medical image visualization with AI-based haptic interaction.

The requirements of data rates and estimated bandwidths for sensors with common specifications and control signals are reported in Table 1 [20]. Specific considerations should be given to the control loops, which could not be performed locally.

**Table 1.** Requirements for sensors and signals.

| Sensor | Specs | Bandwidth |
|---|---|---|
| Cameras | 2×, 640×480, 30 fps, 8/24 bit | 147–441 Mbps uncompressed |
| Microphones | 2×, 44 kHz, 16 bit | 1.4 Mbps |
| Tactile sensors | 4000×, 50 Hz, 8 bit | 1.6 Mbps |
| Control commands | 53 DoF × 2–4 commands, 100 Hz/1 kHz, 16 bit | 170 Kbps |

Network malfunctions such as network delay, jitter, packet loss, and out-of-order packet delivery gravely affect haptic application, similar to every real-time multimedia streaming application. To avoid such types of disorders, the network conditions that are depicted in Table 2 should be satisfied [21]. Taking into consideration these elements, it is noticeable that haptic applications are sensitive to jitter, delay, and update rate, but they are more sensible to packet loss and low bandwidth compared to the requirements that other multimedia applications have. The objective of 5G systems is to achieve the demanded latency and throughput, with ultra-high reliability.

**Table 2.** Network requirements.

| QoS | Haptics | Video | Audio |
|---|---|---|---|
| Jitter (ms) | ≤2 | ≤30 | ≤30 |
| Delay (ms) | ≤50 | ≤400 | ≤150 |
| Data Rate (Kbps) | 512–1024 | 2500–40,000 | 64–128 |
| Packet Loss (%) | ≤10 | ≤1 | ≤1 |
| Update Rate (Hz) | ≥1000 | ≥30 | ≥50 |

The advantages of 3D viewing and haptic interaction are undeniable, and they are regarded as cutting-edge technologies that will find widespread application in the medical industry. The benefits of 3D communication and accurate visualization of the effects that VR and AR technologies could offer will benefit medical work. These technologies are already widely used in a variety of scientific areas, particularly in the industrial area, and undoubtedly in the medical area [22].

The goal of this model is to provide doctors with an optimized evaluation of clinical results by offering an improved clinical image of patients in medical centers and hospitals. Doctors can operate in a more secure environment and make better decisions for their patients' cases. The doctor can see the corresponding clinical data of each patient in higher resolution and quality, such as Ultra High Definition (UHD), thanks to 3D visualization and the use of haptic interfaces. Using a communal virtual 3D display system that provides concurrent real-time visual feedback monitoring collaboration, the doctor can consult with other doctors from different medical centers and hospitals to diagnose and treat various ailments and diseases.

In addition, this scheme allows more close-up views of hidden areas that are not easy to reveal by using the current conventional medical methods. Such a procedure ensures greater accuracy in the identification of potential complications, more effective patient care, quicker return to daily activities, precise clinical outcomes, faster recovery, and fewer complications, allowing patients to safely return home and visit their family sooner. The suggested approach may be employed in both hospitals and medical facilities. Research facilities can adapt it as well, for use in testing environments for potential improvements.

## 6. Conclusions

In conclusion, an enhanced with artificial intelligence and haptic feedback robotic surgery model for the better and quicker discovery of patients' complications has been

proposed. The proposed model is based on the novel technologies of VR, AR, AI, and haptics, providing multiple advantages both to the doctors and to the patients. With the use of such technologies, our proposed approach offers multiple advantages to the medical staff, opening new horizons both in the prevention and treatment of various diseases and in robotic surgery. Thanks to these immersive technologies, surgeons' work can be easier and better with great dexterity, enhanced precision, and real-time visual feedback. In this way, the doctors can have improved 3D imaging health data using haptics on a computer that depicts the clinical results of the patients. On the other hand, the patients enjoy better and quicker medical results with reduced pain and discomfort. The recovery time is reduced significantly, and thus patients can leave the hospital sooner and return back to their normal activities. Moreover, the precise actions of the robotics lead to smaller incisions, resulting in reduced risk of infection and reduced blood loss.

Additionally, 5G enables resource gathering of skilled medical professionals via ultra-reliable and high-performance telemedicine, using the Tactile Internet with haptic feedback. AI improves the system as it gives the opportunity to become independent from human interaction. The integration of these technologies allows examining and healing patients without the existence of physical contact between doctors and patients. This could be proved as a helpful solution for humanity in cases of an epidemic or pandemic. In the upcoming years, when 5G technology is completely finalized, it will introduce unique e-health applications, while enabling an ad hoc composition of healthcare facilities and services by incorporating patients, medical staff, and social workers through its ubiquitous access services.

Finally, the use of haptic interfaces to support telemedicine applications is a very promising open research field. Thus, the deployment of an application where a doctor and a patient are not in the same place could become widely useful. In addition, personalized health consultation will turn out to be available through the advancements in Big Data and Cloud Computing. Furthermore, everyday behaviors and activities of individuals, involving remote health analysis, could be managed by Machine Learning algorithms in order to produce an accurate diagnosis. In this way, the overall healthcare industry paves the route to a very prosperous future, thereby benefiting humanity. Thus, an investment in equipment and upfront development may produce multiple healthcare industry benefits. The introduction of emerging technologies in healthcare may sound complex and costly, but the harvest of the potential advantages will compensate for all these difficulties.

**Author Contributions:** Conceptualization, G.M.M., V.A.M., K.D.S. and C.L.S.; methodology, G.M.M. and V.A.M.; software, G.M.M. and V.A.M.; validation, G.M.M., V.A.M. and K.D.S.; formal analysis, G.M.M. and K.D.S.; investigation, G.M.M., V.A.M. and K.D.S.; resources, G.M.M., V.A.M. and K.D.S.; data curation, G.M.M., V.A.M. and K.D.S.; writing—original draft preparation, G.M.M., V.A.M. and K.D.S.; writing—review and editing, G.M.M., V.A.M., K.D.S. and C.L.S.; visualization, G.M.M., V.A.M., and K.D.S.; supervision, K.E.P.; project administration, K.E.P. All authors have read and agreed to the published version of the manuscript.

**Funding:** This research received no external funding.

**Institutional Review Board Statement:** Not applicable.

**Informed Consent Statement:** Not applicable.

**Data Availability Statement:** Not applicable.

**Conflicts of Interest:** The authors declare no conflict of interest.

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
