# Peer review of "A Medical Image Visualization Technique Assisted with AI-Based Haptic Feedback for Robotic Surgery and Healthcare"

_applsci, doi:10.3390/app13063592_

Round 1
Reviewer 1 Report
This is a review of different modern methods that can help in medicine. It is an interesting text, but it does not bring a clear proposal for a solution, but several solutions are still to be tested.
What is the purpose of the study? It was not defined.
A sentence from l.81-82 despite having reference (2) to the statement seems to be just a loose and overestimated assumption, it is not a scientifically validated statement yet. Rewrite.
A sentence from l.105-107 seems overrated. Add objective values, how long does it take? What percentage of complications are due to this statement? Rewrite.
A phrase from l.109, 111-112. Numerous ? How many? How much is the risk of infections? Costly? How much ? Rewrite.
4. Proposed Approach – The purpose of the study was not clear. It has to be related to the title of the article. You have to make the article about robotics more specific for surgery, digressing from the topic. Also, rewrite the connection of 3D anatomical images with the risk of pandemic diseases and their treatment. The proposal was too broad and exaggerated. This could be spelled out in the discussion, making the objective more concise and specific.
4.1. Virtual Reality (VR) rewrite this part. Direct with the objective of the study.
There is a confusing mix of concepts, mobility problems, reducing pain, treating phobias, post-traumatic stress, studying various diseases in the study of viruses, vaccine research, etc. Explore this in the discussion. What´s the connection of this with the paper title?
A paragraph from l.131-137 was confusing. Rewrite.
4.2. Augmented reality (AR) rewrite in more detail for readers who are new to the topic.
In l.186 better explain what you mean by “ Risky deformities ”. What is the definition of this term? It seems exaggerated to the reader looking for scientific information.
Explain better how “infrared works depth-sensing cameras” and their importance in AR. Don't withdraw, just explain better. Add references.
Check l.205. It is only possible to state that there will be no infection if the surgery is virtual; otherwise, real and with surgical incision, there will always be a risk of infection. Rewrite.
In l.208 you mentioned “nursing staff”, it was not clear why you mixed it with medical support.
In l.221, mixing “mental health and psychotherapy ” with “surgical incisions” was not good. They are very different themes for different types of readerships.
The statement in l.248 is not completely true, as there is a need for a surgeon to introduce the surgical instruments into the abdominal cavity, eg, this is still far from being carried out safely and completely by a robot from the outside. The statement is very exaggerated.
It was unclear what the author meant by “proactive care”, in addition to being a very broad concept.
In l.264, rewriting “simple as well as complex problems”, is redundant.
Paragraph l.274 looks like a transcription of a journalistic report. Rewrite and put scientific references.
In l.288 the author raises another question “ethical concerns”. The text has to be concise and has a clearer development for the reader. It is rambling a lot in several areas of knowledge.
4.6. 5G Networks In this part it is confusing whether to increase 100 or 50 times. It seems that currently, it is 20 times. Check and reference current scientific literature.
Rewrite the sentence: “Experienced surgeons will “transform” into teachers, and amateur ones into students.” This is not what defines a teacher or student.
Do not use exaggerated terms such as “This exhilarating glimpse ”, “empower”, “unprecedented”
What do the authors mean by “decentralization of hospital services”? Rewrite.
The authors bring up other questions “electromagnetic waves straightforwardly affect human lives”. The text has to be concise and has a clearer development for the reader. It is rambling a lot in several areas of knowledge.
In line 239, the sentence is repeated.
L. 332. In various parts the authors mention diagnosis. Make it clearer what type of diagnosis you are referring to nosological, anatomical, functional, anatomopathological, structural, initial, precocious, evolutionary, etc. The concept is very broad.
Statement l.343 is not completely true; there is a possibility of spreading the disease, as care does not depend only on a surgeon.
What is the authors' reference to assemble Tables 1 and 2? There is no clear justification in the text, it does not explain how the authors arrived at these values and not others. It's just a suggestion. This could be the results and the conclusion but must be proved in the paper.
Paragraphs of l.369 are without bibliographic reference.
What procedure are the authors referring to in l.379? To specify.
Paragraph l.383 is just an assumption. No proof or validation has been provided. It's not a result, it's just an idea.
Conclusion. I do not agree that a solution or result has been presented. It's just an assumption without validation. The authors did not do any further experiments or analyses to conclude it. The conclusion is a summary of everything that was proposed, it is not a conclusion by itself, because the objective was not defined. Rewrite.
Author Response
Manuscript Title:
A Medical Image Visualization Technique Assisted with ΑI-based Haptic Feedback for Robotic Surgery and Healthcare
Dear Editor,
Thank you for allowing a resubmission of our manuscript, with an opportunity to address the reviewer’s comments.
Following is our response to the reviewer’s comments.
Best regards,
The authors
Reviewer:
This is a review of different modern methods that can help in medicine. It is an interesting text, but it does not bring a clear proposal for a solution, but several solutions are still to be tested.
Authors’ response: Thank you for the review and your helpful comments.
What is the purpose of the study? It was not defined.
Authors’ response: We have made slight changes in the Introduction by adding the contributions in a bulleted form.
A sentence from l.81-82 despite having reference (2) to the statement seems to be just a loose and overestimated assumption, it is not a scientifically validated statement yet. Rewrite.
Authors’ response: We have rewritten this sentence.
A sentence from l.105-107 seems overrated. Add objective values, how long does it take? What percentage of complications are due to this statement? Rewrite.
Authors’ response: The authors have rewritten the sentence.
A phrase from l.109, 111-112. Numerous ? How many? How much is the risk of infections? Costly? How much ? Rewrite.
Authors’ response: The authors mention the corresponding reference to provide evidence of the written phrase.
- Proposed Approach – The purpose of the study was not clear. It has to be related to the title of the article. You have to make the article about robotics more specific for surgery, digressing from the topic. Also, rewrite the connection of 3D anatomical images with the risk of pandemic diseases and their treatment. The proposal was too broad and exaggerated. This could be spelled out in the discussion, making the objective more concise and specific.
Authors’ response: We have added some text in order to explain concisely the proposed approach.
4.1. Virtual Reality (VR) rewrite this part. Direct with the objective of the study.
Authors’ response: Section 4.1 has been updated to focus more on the objective of our study.
There is a confusing mix of concepts, mobility problems, reducing pain, treating phobias, post-traumatic stress, studying various diseases in the study of viruses, vaccine research, etc. Explore this in the discussion. What´s the connection of this with the paper title?
Authors’ response: Indeed. We have extended the title of the paper to cover all other medical activities.
A paragraph from l.131-137 was confusing. Rewrite.
Authors’ response: We have rewritten this part.
4.2. Augmented reality (AR) rewrite in more detail for readers who are new to the topic.
Authors’ response: The authors have rewritten more comprehensive this topic.
In l.186 better explain what you mean by “Risky deformities ”. What is the definition of this term? It seems exaggerated to the reader looking for scientific information.
Authors’ response: Indeed. We have substitute this term with the most-known term in the medical community “dangerous distortions”.
Explain better how “infrared works depth-sensing cameras” and their importance in AR. Don't withdraw, just explain better. Add references.
Authors’ response: The following references has been added:
- Strickland, J. Tremaine, G. Brigley, and C. Law, “Using a depth-sensing infrared camera system to access and manipulate medical imaging from within the sterile operating field,” Canadian Journal of Surgery. Journal Canadien De Chirurgie, vol. 56, no. 3, pp. E1-6, Jun. 2013.
- Addison, P. S. Addison, P. Smit, D. Jacquel, and U. R. Borg, “Noncontact Respiratory Monitoring Using Depth Sensing Cameras: A Review of Current Literature,” Sensors, vol. 21, no. 4, p. 1135, Feb. 2021.
Check l.205. It is only possible to state that there will be no infection if the surgery is virtual; otherwise, real and with surgical incision, there will always be a risk of infection. Rewrite.
Authors’ response: We have rewritten this paragraph as follows:
“This ability of the surgeon to interact with the patient in such a way is very important to avoid any kind of infection, especially in the case of a completely virtual surgery.”
In l.208 you mentioned “nursing staff”, it was not clear why you mixed it with medical support.
Authors’ response: We have removed this term.
In l.221, mixing “mental health and psychotherapy ” with “surgical incisions” was not good. They are very different themes for different types of readerships.
Authors’ response: Indeed. For this reason, we have removed the sentence that mentioned about mental health and psychotherapy.
The statement in l.248 is not completely true, as there is a need for a surgeon to introduce the surgical instruments into the abdominal cavity, eg, this is still far from being carried out safely and completely by a robot from the outside. The statement is very exaggerated.
Authors’ response: We have rewrite this sentence as follows:
“Thus, haptic data reinforced by low latency networks will make available VR-based headsets that can assist surgeons and doctors in executing partially operations via telepresence, providing many benefits.”
It was unclear what the author meant by “proactive care”, in addition to being a very broad concept.
Authors’ response: For this reason, we have removed the sentence with this term.
In l.264, rewriting “simple as well as complex problems”, is redundant.
Authors’ response: We have removed this sentence.
Paragraph l.274 looks like a transcription of a journalistic report. Rewrite and put scientific references.
Authors’ response: We have added the following references in this paragraph:
- Jordan, M. I. (2019). Artificial intelligence—the revolution hasn’t happened yet. Harvard Data Science Review, 1(1), 1-9.
- Jobard, M.-L. Brandely-Piat, F. Chast, and R. Batista, “Qualification of a chemotherapy-compounding robot,” Journal of Oncology Pharmacy Practice: Official Publication of the International Society of Oncology Pharmacy Practitioners, vol. 26, no. 2, pp. 312–324, Mar. 2020.
In l.288 the author raises another question “ethical concerns”. The text has to be concise and has a clearer development for the reader. It is rambling a lot in several areas of knowledge.
Authors’ response: Indeed. For this reason, we have removed this sentence.
4.6. 5G Networks In this part it is confusing whether to increase 100 or 50 times. It seems that currently, it is 20 times. Check and reference current scientific literature.
Authors’ response: We have corrected the typo.
Rewrite the sentence: “Experienced surgeons will “transform” into teachers, and amateur ones into students.” This is not what defines a teacher or student.
Authors’ response: We have rewritten this sentence.
Do not use exaggerated terms such as “This exhilarating glimpse ”, “empower”, “unprecedented”
Authors’ response: We have altered these terms.
What do the authors mean by “decentralization of hospital services”? Rewrite.
Authors’ response: The authors have rewritten more comprehensive this sentence.
The authors bring up other questions “electromagnetic waves straightforwardly affect human lives”. The text has to be concise and has a clearer development for the reader. It is rambling a lot in several areas of knowledge.
Authors’ response: The authors have rewritten this part in order to become clearer to the reader.
In line 239, the sentence is repeated.
Authors’ response: The text has been corrected.
- 332. In various parts the authors mention diagnosis. Make it clearer what type of diagnosis you are referring to nosological, anatomical, functional, anatomopathological, structural, initial, precocious, evolutionary, etc. The concept is very broad.
Authors’ response: Explanation of text was made.
Statement l.343 is not completely true; there is a possibility of spreading the disease, as care does not depend only on a surgeon.
Authors’ response: The appropriate explanation has been given.
What is the authors' reference to assemble Tables 1 and 2? There is no clear justification in the text, it does not explain how the authors arrived at these values and not others. It's just a suggestion. This could be the results and the conclusion but must be proved in the paper.
Authors’ response: The appropriate citation has been given.
Paragraphs of l.369 are without bibliographic reference.
Authors’ response: We have added the corresponding reference
What procedure are the authors referring to in l.379? To specify.
Authors’ response: The appropriate explanation has been given.
Paragraph l.383 is just an assumption. No proof or validation has been provided. It's not a result, it's just an idea.
Authors’ response: Explanation of text was made.
Conclusion. I do not agree that a solution or result has been presented. It's just an assumption without validation. The authors did not do any further experiments or analyses to conclude it. The conclusion is a summary of everything that was proposed, it is not a conclusion by itself, because the objective was not defined. Rewrite.
Authors’ response: The authors referred the main advantages of such an approach.

Reviewer 2 Report
The authors made a nice attempt use various industry 4.0 technologies (AR,VR,MR with AI, 5G and Robotics) in the health care to handle critical situations like pandemics. More specifically the challenges in communication and visualisations have been presented. The manuscript shall be further improved with opportunities in the related subject matter.
Numerous literatures is available on these keywords, it is recommended to expand section 2 “Related Work”
While discussing the challenges in AR/VR/MR, I suggest some literature for further consideration, where the authors get some insights into further opportunities/options. In addition to the suggested, the authors can also see some literature with the keywords “Challenges and opportunities in human robot collaboration”
Kolecki, Radek, et al. "Assessment of the utility of mixed reality in medical education." Translational Research in Anatomy (2022): 100214.
Eswaran, M., and MVA Raju Bahubalendruni. "Challenges and opportunities on AR/VR technologies for manufacturing systems in the context of industry 4.0: A state of the art review." Journal of Manufacturing Systems 65 (2022): 260-278.
Figure 1 needs to be modified (AR and AV are part of XR, not VR); please do verify.
In Figure 2, eliminate the wor “Slave” (the authors can use Lead & Follower) instead of (Master & slave)
Expand the conclusion with future scope in terms of technical requirements.
Overall manuscript is well articulated.
Author Response
Manuscript Title:
A Medical Image Visualization Technique Assisted with ΑI-based Haptic Feedback for Robotic Surgery and Healthcare
Dear Editor,
Thank you for allowing a resubmission of our manuscript, with an opportunity to address the reviewer’s comments.
Following is our response to the reviewer’s comments.
Best regards,
The authors
Reviewer:
The authors made a nice attempt use various industry 4.0 technologies (AR, VR, MR with AI, 5G and Robotics) in the health care to handle critical situations like pandemics. More specifically the challenges in communication and visualisations have been presented. The manuscript shall be further improved with opportunities in the related subject matter.
Authors’ response: Thank you for the review and your helpful comments.
Numerous literatures is available on these keywords, it is recommended to expand section 2 “Related Work”
While discussing the challenges in AR/VR/MR, I suggest some literature for further consideration, where the authors get some insights into further opportunities/options. In addition to the suggested, the authors can also see some literature with the keywords “Challenges and opportunities in human robot collaboration”
Kolecki, Radek, et al. "Assessment of the utility of mixed reality in medical education." Translational Research in Anatomy (2022): 100214.
Eswaran, M., and MVA Raju Bahubalendruni. "Challenges and opportunities on AR/VR technologies for manufacturing systems in the context of industry 4.0: A state of the art review." Journal of Manufacturing Systems 65 (2022): 260-278.
Authors’ response: We have expanded the Related Work section by adding the aforementioned literature.
Figure 1 needs to be modified (AR and AV are part of XR, not VR); please do verify.
Authors’ response: We have modified the figure.
In Figure 2, eliminate the word “Slave” (the authors can use Lead & Follower) instead of (Master & slave)
Authors’ response: We have modified the figure.
Expand the conclusion with future scope in terms of technical requirements.
Authors’ response: Extension of text was made.
Overall manuscript is well articulated.
Authors’ response: Thank you.

Round 2
Reviewer 1 Report
Dear,
The manuscript submitted for review has undergone significant improvements and is now ready for publication. The revisions made to the manuscript have ensured a clear and comprehensive presentation of the research results, while also enhancing the work's relevance and scientific solidity, making it of great interest to the readers of the publication.
Several changes were made to the manuscript throughout the revision process to ensure readers effectively communicated and understood the research results. Additionally, new information was incorporated, significantly contributing to the comprehension of the results. As a result, the work now stands on a solid theoretical and methodological foundation, increasing its scientific relevance.
I wish to emphasize that the improved manuscript is comprehensive, highly relevant, and of great interest to the publication's readers. The publication of this work will undoubtedly contribute significantly to the advancement of knowledge in the relevant field.
Thank you for providing the opportunity to review this manuscript. I look forward to its publication in the journal.
Yours sincerely,